# Comprehensive Landscape of RRM2 with Immune Infiltration in Pan-Cancer

**DOI:** 10.3390/cancers14122938

**Published:** 2022-06-14

**Authors:** Zijian Zhou, Qiang Song, Yuanyuan Yang, Lujia Wang, Zhong Wu

**Affiliations:** 1Department of Urology, Huashan Hospital, Fudan University, Shanghai 200040, China; zjzhou21@m.fudan.edu.cn (Z.Z.); yyyang20@fudan.edu.cn (Y.Y.); 2Institute of Urology, Fudan University, Shanghai 200040, China; 3Department of Urology, The First Affiliated Hospital of Nanjing Medical University, Nanjing 210029, China; qiangsong121@163.com

**Keywords:** RRM2, pan-cancer, prognosis, immune infiltration, immunotherapy, bladder cancer

## Abstract

**Simple Summary:**

RRM2 is a crucial subunit of ribonucleotide reductase. In this article, we provided a comprehensive analysis of RRM2 with immune infiltration in pan-cancer. We focused on the hotspots of ferroptosis-related gene RRM2 and immunotherapy. Via bioinformatics analysis, multiple indicators suggested that RRM2 high expression may enhance immunotherapy sensitivity. For the first time, we systematically analyzed the role of RRM2 in pan-cancer. We provided the prospect of RRM2 and immunotherapy for pan-cancer. Additionally, we proved the expression pattern, clinical value, prognostic value and potential pathways of RRM2 with different platforms. In particular, we confirmed RRM2 expression and function in bladder cancer in our clinical samples and cell lines. Collectively, we found that RRM2 is a novel prognostic biomarker, and these findings may aid in an improved understanding of the role of RRM2 and its clinical application in human cancers.

**Abstract:**

As a crucial subunit of ribonucleotide reductase, RRM2 plays a significant part in DNA synthesis. This study aimed to elucidate the comprehensive landscape of RRM2 in human cancers. With different bioinformatics platforms, we investigated the expression pattern, prognostic significance, mutational landscapes, gene interaction network, signaling pathways and immune infiltration of RRM2 in tumors. We found that RRM2 expression was predominantly up-expressed in tumor tissues in most tumors. Concurrently, RRM2 expression was significantly associated with worse prognosis and tumor stage across TCGA cancers. Moreover, RRM2 high levels were critically associated with the infiltration of natural killer T cells and immune scores. RRM2 was positively related to immune checkpoints, tumor mutation burden, microsatellite instability, neoantigen, and cytotoxic T lymphocyte in several cancers, predicting effective response to immunotherapy. Meanwhile, a strong co-expression of RRM2 with immune-related genes was observed. Additionally, multiple Cox regression analysis showed that RRM2 was an independent prognostic factor in bladder cancer (BLCA). Eventually, we verified that RRM2 was overexpressed in BLCA clinical samples and cell lines. Blocking RRM2 could suppress BLCA cells’ growth and proliferation while enhancing sensitivity to cisplatin. This study provided a new perspective for understanding RRM2 in cancers and new strategies for tumor immunotherapy.

## 1. Introduction

Cancer therapy has advanced substantially over the past decades, yet many patients still undergo cancer progression, recurrence and, ultimately, death [1]. In recent years, antitumor immunotherapy has provided considerable survival benefits in the treatment of multiple tumors, such as melanoma, colorectal cancer, and lung cancer [2,3]. The discovery of tumor immune escape mechanisms promotes the development of immune checkpoint inhibitors (ICIs) [4]. However, non-sensitivity and adverse reactions to ICIs limited their practical application [5,6]. Therefore, there is a need to elucidate the mechanisms of cancer pathogenesis to predict and increase the sensitivity of ICIs for tumor patients. The continuously improving cognition of tumor genomic characterization and the immune microenvironment has major implications for finding biomarkers to guide personalized immunotherapy [7,8].

Ribonucleotide reductase (RR) is the rate-limiting enzyme that catalyzes the de novo formation of deoxyribonucleotides in DNA synthesis [9]. RR mainly consists of two homodimeric subunits—the large subunit (RRM1) and the small subunit (the small subunit has two isoforms, RRM2 and RRM2B) [10]. As the critical component of RR, RRM2 contains tyrosyl radicals necessary to initiate catalysis and participates in the regulation and modification of proteins [11,12]. RRM2 aberration may lead to genome instability and increased mutation rates, thus influencing tumor progression [13,14,15]. For instance, RRM2 could significantly and specifically promote colon cancer (CRC) cell growth [16]. Consequently, RRM2 overexpression plays a critical part in the proliferation, metastasis and drug dependence of human cancers [17,18].

More importantly, RRM2 has been identified as a member of the ferric iron-binding ferritin-superfamily [19]. Shen et al. have demonstrated that RRM2 engaged in the iron metabolism in hepatocellular carcinoma (HCC) [20]. Iron modulated tumor cells progress through protein and lipid modifications [21]. However, iron overload also contributed to cancer cell death. This accumulation of iron-dependent lipid peroxides leads to a new type of cell death, known as ferroptosis [22,23]. As an example, Yang et al. found that RRM2 could protect against ferroptosis by sustaining glutathione (GSH) synthesis in liver cancer cells [24]. Therefore, despite the intensive efforts to investigate RRM2 function in cancers before, the brand-new role of RRM2 in ferroptosis and iron homeostasis is still worth pursuing. Moreover, the composite function of RRM2 cooperating with immune cells, and the molecular mechanisms and detailed pathways of RRM2 in human cancers remain enigmatic. 

Given these limitations, we conducted an integrative bioinformatics analysis of RRM2 in pan-cancer. At first, we investigated the expression patterns of RRM2 and evaluated the prognostic values of RRM2 in pan-cancer with public databases. Next, we explored the key genes and signal pathways regulated by RRM2. Notably, we identified the relation of RRM2 to the tumor microenvironment (TME), ICIs and researched its regulation with immune infiltration and immunotherapy. Finally, we clarified the protein expression of RRM2 in pan-cancer and researched the biological functions of RRM2 in bladder cancer (BLCA) in vitro. These observations could be of significance for basic research as well as clinical application and may improve precision for cancer immunotherapy.

## 2. Materials and Methods

### 2.1. Clinical Samples 

A total of 32 matched bladder tumor and adjacent normal specimens were obtained by radical nephrectomy in Huashan Hospital, Fudan University. All diagnosis was confirmed by histopathological examination. The specimens were frozen in liquid nitrogen after surgery for further research. All 32 patients had given written informed consent and a normalized ethnic audit has been conducted. 

### 2.2. Cell Culture, Quantitative Real-Time PCR (qRT-PCR) and Western Blot Assays

BLCA cell lines (T24, J82, UMUC3, RT4, 5637 and 253J) were obtained from the Chinese Academy of Sciences (Shanghai, China). Simian Virus 40-immortalized human uroepithelial cell (SVHUC) is the human normal urinary epithelial cell line, which is used as a normal control in our study. RRM2 expression was detected by qRT-PCR and western blot assays, respectively. Notably, qRT-PCR was carried out in triplicate with three independent experiments to evaluate RRM2 expression in BLCA cell lines and clinical specimens. The reagents and consumables are listed in Appendix A. 

### 2.3. Cell Transfection, Cell Proliferation, Colony Formation Assays and IC50 Determination

RRM2 siRNA (Hanbio Biotechnology, Nanjing, China) was transfected T24 and UMUC3 cells with Invitrogen Lipofectamine 3000 kit. The cell proliferation was measured after 24 h, 48 h, 72 h and 96 h by cell counting Kit-8 (CCK-8) at 450 nm following incubation at 37 for 1 h. Colony formation assays were performed by seeding 500 cells/well into 6-well plates. After incubating for 1–2 weeks, the formed spheres were counted and photographed. 

### 2.4. RRM2 Data Acquisition and Processing 

RRM2 data and related information of 33 cancer types were collected from The Cancer Genome Atlas (TCGA, https://portal.gdc.cancer.gov/, accessed on 25 February 2022), including 10363 tumor tissues and corresponding paired 730 para-carcinoma tissues. Meanwhile, the normal control gene profiles of normal tissues were downloaded from the GTEx datasets (https://www.gtexportal.org/, accessed on 25 February 2022). RRM2 expression in TCGA and GTEx datasets was analyzed with transcripts per million (TPM) using the same sequencing platform and library preparation to reduce potential batch effects. TCGA and GTEx data were utilized to analyze differential RRM2 mRNA expression between normal and tumor samples using the “beeswarm” and “ggpubr” R packages. Two sets of t-tests were run and *p* < 0.05 indicated differential expression between tumor and normal tissues. Microarray data were downloaded from the GEO database (http://www.ncbi.nih.gov/geo/, accessed on 10 March 2022) and normalized by log2 transformation to perform RRM2 expression analysis. 

### 2.5. Analysis of RRM2 Expression Profiles with Different Public Platforms

The protein expression profiles of RRM2 were explored in the Human Protein Atlas (HPA, https://www.proteinatlas.org/, accessed on 25 February 2022) and UALCAN (http://ualcan.path.uab.edu/, accessed on 25 February 2022). IHC images of RRM2 (antibody HPA056994) were obtained from HPA. As a web-based tool, UALCAN analyzes transcriptome data from TCGA and MET500 data to confirm the protein expression of RRM2 in pan-cancer. 

### 2.6. Survival Analysis and Cox Regression Analyses

Kaplan-Meier (KM) curves were analyzed with TCGA datasets. The median of RRM2 expression was used as a cutoff value. The log-rank *p*-value and Cox *p*-value with a hazard ratio (HR) were calculated to assess the prognostic value of RRM2, including overall survival (OS), disease-specific survival (DSS), progression-free survival (PFS) and disease-free survival (DFS). The “survival” and “forest plot” R packages were utilized to visualize the survival analysis. Accordingly, the prognostic value of RRM2 was confirmed in Kaplan-Meier Plotter (KM Plotter, http://kmplot.com/, accessed on 26 February 2022) and PrognoScan (http://dna00.bio.kyutech.ac.jp/PrognoScan/index.html/, accessed on 26 February 2022). 

### 2.7. RRM2 Genomic Alterations Analysis

Analysis of RRM2 genomic alterations among TCGA was calculated in cBioPortal database (http://www.cbioportal.org/, accessed on 26 February 2022). The cBioPortal for Cancer Genomics provides access to a large-scale cancer genomics dataset that can be visualized, downloaded and analyzed. RRM2 genomic alterations included alteration frequency, copy number alterations and mutations. In the “OncoPrint” and “Cancer Types Summary” modules, 10953 cancer patients were analyzed for RRM2 genomic alteration types and alteration frequency. Tumor mutation burden (TMB), microsatellite instability (MSI) and neoantigen (NEO) scores were available from UCSC XENA database and their association with RRM2 expression was explored with Spearman’s correlation analysis. 

### 2.8. RRM2-Interacting Genes and Protein-Protein Interaction (PPI) Network

GeneMANIA database (http://www.genemania.org/, accessed on 26 February 2022) was applied to construct the gene-gene interaction network of RRM2. GeneMANIA is an interactive and user-friendly website for building a gene-gene interaction network, which provides gene function prediction hypotheses and identifies genes with comparable roles. STRING database (https://string-db.org/, accessed on 26 February 2022) was utilized to build a PPI network of RRM2 for construction, visualization and analysis. To generate the PPI network, the Search Tool for the Retrieval of Interacting Genes (STRING) was applied with the following input parameters: “evidence”, “experiments” and a 0.700 confidence level. The first 100 RRM2-correlated genes with *p* < 0.05 were acquired from GEPIA2.0 (http://gepia2.cancer-pku.cn/#index, accessed on 26 February 2022) database with its “Similar Genes Detection” function. GEPIA is a user-friendly web portal for gene expression analysis based on TCGA and GTEx data. In the current study, the “Correlation Analysis” module of GEPIA2 was applied to compute pair-wise gene expression correlations between RRM2 and the selected top 5 genes using the Pearson correlation method. Then with TIMER2.0 database (http://timer.cistrome.org/, accessed on 26 February 2022), the heatmap of the top 5 genes (CCNA2, CKAP2L, KIF11, MKI67 and PLK1) was acquired, containing the partial correlation coefficient (cor) and *p*-value calculated by the purity-adjusted Spearman’s rank correlation test. Ultimately, the RRM2 cooperated and associated gene, RRM1, was selected by the intersection analysis with a Venn diagram. 

### 2.9. Functional Enrichment Analysis 

Gene Ontology (GO) and Kyoto Encyclopedia of Genes and Genomes (KEGG) analyses were performed to research the possible biological functions and signaling pathways of RRM2. GO analysis is a powerful bioinformatics tool for determining RRM2′ biological processes (BPs), cellular components (CCs) and molecular functions (MFs). KEGG enrichment analysis was conducted by the Metascape portal with a comprehensive gene list annotation and analysis resource for experimental biologists. The resulting enriched pathways were visualized using the “ggplot2” R package. The single-gene Gene Set Enrichment Analysis (GSEA) analysis of RRM2 was applied to investigate the potential signal paths. The genes correlated with RRM2 (*p* < 0.05) were ranked and subjected to GSEA analysis. The top 15 terms of GSEA from Reactome pathway were exhibited with adjust *p* < 0.05 by “clusterProfiler” R package. 

### 2.10. Immune Infiltration and Immune-related Genes Analysis 

Based on RRM2 expression data, the “estimate” R package was used to explore the abundance of estimated stromal and immune cells (StromalScore and ImmuneScore). Then the composition of immune cell infiltration was assessed by the “GSVA” R package with Spearman’s rho value. 

Next, the XCELL algorithms were utilized to estimate the Natural killer T (NKT) cells infiltration based on TIMER2.0. Spearman correlations between RRM2 and immune cell markers were investigated in BLCA via TIMER. Then we collected information on RRM2 and immunomodulators, including 46 immune stimulators, 21 major histocompatibility complex (MHC), 41 chemokines and 18 receptors. The immunotherapeutic response outcomes of RRM2 in melanoma immunotherapy cohorts (Riza2017_PD1) and the correlation of RRM2 with CTL were analyzed based on Tumor Immune Dysfunction and Exclusion database (TIDE, http://tide.dfci.harvard.edu/, accessed on 5 March 2022)

### 2.11. Statistical Analysis 

R software (version 4.0.1), SPSS (version 26.0) and GraphPad Prism (version 8.0) were utilized to perform the statistical analysis. The rational statistical test was employed to compare two independent test series (Student’s t-test) or more test series (ANOVA test). Data were shown as average values ± SEM and a two-sided *p* < 0.05 indicated statistical significance.

## 3. Results

### 3.1. Overexpression of RRM2 in Pan-Cancer 

RRM2 mRNA expression was first assessed in pan-cancers. High RRM2 expression was observed in 19 cancers via TCGA: BLCA, BRCA, CESC, CHOL, COAD, ESCA, GBM, HNSC, KIRC, KIRP, LIHC, LUAD, LUSC, PCPG, PRAD, READ, STAD, THCA and UCEC (Figure 1A). Consistently, overexpression of RRM2 was discovered in 28 cancers via GTEx-TCGA: ACC, BLCA, BRCA, CESC, CHOL, COAD, ESCA, GBM, HNSC, KIRC, KIRP, LAML, LIHC, LGG, LUAD, LUSC, OV, PAAD, PCPG, PRAD, READ, SKCM, STAD, TGCT, THYM, THCA, UCS and UCEC (Figure 1B). Compared with paired para-cancerous samples in TCGA, RRM2 expression also increased in the paired pan-cancer samples of 16 types of cancers: BLCA, BRCA, CHOL, COAD, ESCA, HNSC, KIRC, KIRP, LIHC, LUAD, LUSC, PRAD, READ, STAD, THCA, and UCEC (Figure 1C). 

Moreover, RRM2 protein expression was evaluated in CTPAC samples with UALCAN. RRM2 protein expression increased in BRCA, COAD, OV, KIRC, UCEC, LUSC, LUAD, PDAC, HNSC, GBM and LIHC (Figure 1D). Next, we confirmed RRM2 expression in GEO. RRM2 mRNA expression was elevated in the following 20 types of cancers in corresponding data sets, including ACC, BLCA, BRCA, CESC, COAD, CHOL, ESCA, GBM, HNSC, KIRC, KIRP, LIHC, LUAD, OV, PAAD, PRAD, SARC, SKCM, STAD and UCEC (Figure 1E). Then with IHC images from HPA database, we showed protein levels of RRM2 in several cancers. Stronger staining of RRM2 was detected in BLCA, COAD, LIHC and PDAC tissues than in corresponding normal tissues (Figure 2A). These findings substantiate that upregulated RRM2 might be a paramount culprit in human cancers.

### 3.2. RRM2 Correlates with Tumor Stages and Prognosis in Pan-Cancer

We assessed the correlation of RRM2 with tumor stages in TCGA. As shown in Figure 2B, RRM2 expression was significantly related to tumor stage in ACC, BRCA, COAD, KICH, KIRC, KIRP, LIHC, LUAD, LUSC, SKCM, THCA and OV. Remarkably, RRM2 overexpression leads to advanced tumor stages in KICH, KIRP and LUAD. Additionally, ROC analysis illustrated that RRM2 had a good prediction accuracy (AUC > 0.90) for BRCA, CHOL, COAD, ESAD, ESCA, KIRC, LIHC, LUAD, LUSC, OSCC, STAD and UCEC (Appendix A). These findings suggest that RRM2 expression is correlated with tumor stages in human cancers.

For assessing the prognostic value of RRM2, we estimated the OS with the univariate Cox regression analysis in TCGA (Figure 3A). The results showed that RRM2 overexpression was significantly related to poor OS in ACC (HR = 4.30, *p* < 0.001), BLCA (HR = 1.45, *p* = 0.014), KICH (HR = 8.20, *p* = 0.047), KIRC (HR = 1.97, *p* < 0.001), KIRP (HR = 3.29, *p* < 0.001), LGG (HR = 2.13, *p* < 0.001), PAAD (HR = 1.83, *p* = 0.004), PRAD (HR = 5.11, *p* = 0.046), UCEC (HR = 1.60, *p* = 0.027), UVM (HR = 5.33, *p* = 0.004). Only in THYM (HR = 0.17, *p* = 0.031) did RRM2 high expression predict better OS. KM survival curves were also constructed to confirm the prognostic value of RRM2. Consistently, the results prompt that RRM2 high expression could lead to worsen OS in the above-mentioned cancers, except in THYM (Figure 3B–O). 

DSS survival contributions of RRM2 in pan-cancer were exhibited in Figure 4A. RRM2 high transcriptional levels were also correlated with bad DSS in 10 types cancers: ACC (HR = 4.55, *p* < 0.001), BLCA (HR = 1.64, *p* = 0.007), KIRC (HR = 3.01, *p* < 0.001), KIRP (HR = 29.70, *p* < 0.001), LGG (HR = 2.19, *p* < 0.001), LIHC (HR = 2.04, *p* = 0.002), LUAD (HR = 1.81, *p* = 0.002), MESO (HR = 3.73, *p* < 0.001), PAAD (HR = 1.72, *p* = 0.022) and UVM (HR = 8.39, *p* < 0.001). Coherently, KM analysis with a log-rank test also indicated RRM2 high expression was clearly correlated with worse DSS (Figure 4B–I). Moreover, we explored the effect of RRM2 expression on PFS and DFS (Appendix A). The results of forest plot suggested that RRM2 high expression predicted poor PFS in ACC, BRCA, KICH, KIRC, KIRP, LGG, LIHC, LUAD, MESO, PAAD, PRAD, SARC, THCA and UVM, but indicated good PFS in COAD (Appendix A). Meanwhile, a significant association between RRM2 expression and adverse DFS was observed in KIRP, LUAD, PAAD, SARC, TGCT and THCA (Appendix A). 

Furthermore, we validated the prognostic value of RRM2 in PrognoScan and KM Plotter databases. RRM2 high expression was linked to a dismal OS in several cancers from GEO (Figure 4J), including BRCA (HR = 2.41, *p* < 0.001), GBM (HR = 1.94, *p* < 0.001), LUAD (HR = 1.97, *p* < 0.001), LUSC (HR = 1.54, *p* = 0.047), PRAD (HR = 1.34, *p* < 0.001) and SKCM (HR = 3.46, *p* = 0.002). Similarly, RRM2 high expression influenced DSS in BRCA (HR = 3.23, *p* < 0.001) and COAD (HR = 0.062, *p* = 0.017). Likewise, the results from KM Plotter showed that RRM2 high expression portends poor OS in BLAC, ESCA, KIRP, KIRC, LUAD, LIHC, PDAC, SARC, STAD, READ, THYM and UCEC (Appendix A). These results manifest that RRM2 shows promise as a new prognostic marker for cancer patients and the relevance of RRM2 to tumor stages may help uncover the new underlying mechanisms of tumors.

### 3.3. Genetic Alteration and Mutation Landscape of RRM2 in Cancers

Genetic and epigenetic alterations may induce changes in gene expression and they are closely associated with tumorigenesis and progression. Genetic alterations of RRM2 in pan-cancer samples were analyzed in cBioPortal. High RRM2 alteration frequencies occurred in UCS, UCEC and BLCA (Appendix A). Besides, we observed that all genetic alteration cases of DLBC and KICH carried RRM2 gene deep deletions while ESAD and MESO harbored amplifications. The 3D structure of the RRM2 protein was shown in Appendix A. The genome sites, types and numbers of RRM2 genomic alterations were depicted in Appendix A. Missense mutation of RRM2 is the primary type of RRM2 mutation and the A128V/T alteration in the Ribonuc_red_sm domain is detected in four cases. The general mutation count of RRM2 was presented in Appendix A. Gene alteration analyses of RRM2 should therefore provide further insights into the roles of RRM2 in cancer progression.

### 3.4. Network of RRM2-Interacting Genes and Enrichment Analysis of RRM2-Related Partners 

For pathway enrichment analyses, we investigated the RRM2 cooperated and correlated genes. We acquired the top 100 RRM2-associated genes in pan-cancer with the GEPIA2 database. The correlation heatmap showed the top five genes (CCNA2, CKAP2L, KIF11, MKI67 and PLK1) that were positively and significantly related to RRM2 in almost all TCGA cancers (Figure 5A). As seen in Figure 5B, RRM2 expression was positively related to CCNA2 expression (R = 0.76, *p* < 0.001), CKAP2L (R = 0.74, *p* < 0.001), KIF11 (R = 0.75, *p* < 0.001), MKI67 (R = 0.78, *p* < 0.001) and PLK1 (R = 0.74, *p* < 0.001). Then, we utilized GeneMania and STRING databases for RRM2 to create the PPI network. The results depicted that the top 20 potential target genes interacted with RRM2, including RRM1, RRM2B, KIF11 and so on (Figure 5C,D). The common gene, RRM1, was picked via Venn diagram analysis among the three datasets (GEPIA2.0, GeneMania and STRING) and identified (Figure 5E). Next, cooperated genes in GeneMania and STRING were combined to perform GO and KEGG analyses (Figure 5F) and several pathways like “cell cycle”, “p53 signaling pathway”, “GSH metabolism” and “drug metabolism” were enriched in KEGG analysis, hinting that RRM2 may engage in tumor progression, as well as ferroptosis and drug resistance.

GSEA analysis revealed that RRM2 was involved in immune-related signaling pathways in diverse cancers, especially for the antigen that activates the B cell receptor (BCR) leading to generation of second messengers, CD22 mediated BCR regulation, FCGR3A mediated IL10 synthesis, creation of C4 and C2 activators, initial trigging of complement, binding and uptake of ligands by scavenger receptors, signaling by the BCR, complement cascade, etc. (Figure 6A–F). These outcomes depict that RRM2 possibly has a major impact on the tumor immune microenvironment (TIME).

### 3.5. RRM2 Expression Correlates with Immune cells and Tumor Immune Infiltration

We estimated the immune and stromal components via ESTIMATE algorithms and assessed the association between RRM2 and immune infiltrating via CIBERSORT algorithms. Notably, among the 22 immune cells, the correlation matrix portrayed that RRM2 expression had a negative association with CD8+ T cells in various cancers, including AA, BRCA, ESCA, LAML, PAAD, PRAD, SKCM and THCA (Figure 7A). In parallel, RRM2 expression had a positive and significant association with Tregs in KICH, KIRC, KIRP, LIHC, PCPG, THCA and THYM (Figure 7A). Next, as Figure 7B indicated, compared to patients with RRM2 low expression, RRM2 high expression presented higher immune and stromal scores in KIRC and THCA (*p* < 0.001) but indicated lower immune and stromal scores in CESC, ESCA, GBM, LUSC, SARC and UCEC (*p* < 0.05, Figure 7B). In addition, the correlation coefficients landscape calculated by XCELL algorithms showed that RRM2 expression had a negative and significant association with the NKT cells infiltration in the majority of cancers (Figure 7C). Figure 7D listed the representative scattergrams and the results showed that RRM2 expression was negatively related to the NKT cells infiltration in BLCA (Cor = −0.341, *p* = 1.82 × 10^−11^), BRCA (Cor = −0.227, *p* = 4.85× 10^−13^), GBM (Cor = −0.293, *p* = 5.17 × 10^−4^), PRAD (Cor = −0.258, *p* = 9.04 × 10^−8^), LUAD (Cor = −0.266, *p* = 1.85 × 10^−9^), and UVM (Cor = −0.581, *p* = 3.09 × 10^−8^). The profiles illustrate that RRM2 is putatively involved in tumor immune infiltration and functions critically in the immune-oncological interactions.

### 3.6. The Relationships between RRM2 Expression and ICIs, TMB, MSI, NEO and CTL

ICIs participate in the immunosuppressive mechanism influencing the outcome of immunotherapy, so we assessed the correlations between RRM2 expression and ICIs, including CD274, CTLA4, HAVCR2, LAG3, PDCD1, PDCD1LG2, TIGIT and SIGLEC15. Our discoveries showed that RRM2 expression had a close and positive association with almost all ICIs in BLCA, BRCA, HNSC, KIRC, LGG, LIHC, LUAD, OV and THCA, indicating that RRM2 might enhance immunotherapy sensitivity in the above tumors (Figure 8A). TMB, MSI and NEO have been considered to have excellent predictive value for immunotherapy. In general, RRM2 was positively related to TMB in 16 types of cancers: ACC, BLCA, BRCA, COAD, CHOL, KICH, KIRC, LGG, LUAD, LUSC, PAAD, PRAD, SARC, SKCM, STAD and UCEC (*p* < 0.05, Figure 8B). Similarly, the positive correlation between RRM2 expression and MSI achieved significance (*p* < 0.05) in five types of cancer, including COAD, LIHC, TGCT, UCS and UCEC, but RRM2 was negatively related to MSI only in STAD and DLBC (Figure 8C). As Figure 8D showed, seven TCGA cohorts had positive associations with NEO, including BRCA, LGG, LUAD, LUSC, PRAD, STAD and UCEC.

Additionally, we found that patients with RRM2 high expression achieved clinical benefits of PD-1 immunotherapy in melanoma and hence exhibited prolonged OS and PFS in the Riza2017_PD1 clinical study (Figure 8E). Consistently, RRM2 elevated expression was positively related to the level of CTL in BLCA, BRCA, COAD, HNSC, KIRC, LIHC, LUAD, OV and UVM, while being negatively related to LAML (Figure 8F). Our results imply that RRM2 might influence the efficacy of cancer immunotherapy.

### 3.7. Relationship between RRM2 Expression with Immune-Related Genes

The association of RRM2 expression with immune-related genes in cancers showed that RRM2 had a positive correlation with most chemokines, chemokine receptors, MHC genes, and immunostimulatory genes across TCGA cancer types (Figure 9). Distinctly positive associations between RRM2 expression and human leukocyte antigen (HLA)-I and II molecular were observed in several tumors (Figure 9B). In these immunostimulatory marker genes, CD276, MICB, NT5E, PVR and ULBP1 had a significantly positive correlation with RRM2 expression in most tumor types (Figure 9C). In addition, we observed that RRM2 expression was positively correlated with chemokines and chemokine receptors, such as CCL5, CXCL6 and their receptors CCR5 and CXCR6 (Figure 9A,D). 

### 3.8. Validation of RRM2 Expression and Function in BLCA

Even more importantly, we calculated the correlations of RRM2 with clinicopathological parameters in TCGA-BLCA. Logistic regression analysis deciphered that RRM2 expression was correlated with tumor grade (*p* = 0.003) while it was not correlated with tumor stage (*p* = 0.207) in BLCA (Table 1). Then we validated the prognostic role of RRM2 BLCA samples with multivariable Cox regression. We found that RRM2 was an independent prognostic factor of OS (HR = 1.693, *p* = 0.030) and DSS (HR = 2.482, *p* = 0.004) in TCGA-BLCA (Table 2). Moreover, RRM2 was correlated with diverse immune cell markers in BLCA (Appendix A), whereas a more real investigation of immune infiltrates in BLCA based on IHC remains to explore.

In bioinformatics analysis, we found RRM2 was significantly differentially expressed and predicted poor survival in BLCA. Meanwhile, RRM2 was related to immune infiltration and immunotherapy in BLCA, so we assumed that RRM2 might function in BLCA. Thus, we further confirmed the expression and function of RRM2 in BLCA through experiments. In concurrence with the results of bioinformatic prediction, the RRM2 mRNA level was also elevated in BLCA tumor tissues by qRT-PCR (Figure 10A). Additionally, qRT-PCR results displayed that RRM2 expression was elevated in BLCA cell lines (T24, J82, UMUC3, RT4, 5637 and 253J) relative to SVHUC (Figure 10B). As presented in Figure 10C-E, qRT-PCR and western bolt results showed that RRM2 siRNA was transfected in T24 and UMUC3 cells successfully. The original uncropped western blots were shown in Appendix A. CCK-8 assays showed that blocking RRM2 could significantly inhibit BLCA cell growth and proliferation (Figure 10F). Besides, after suppressing RRM2 expression, the tumor inhibition rate was obviously increased after 48h-treatment with different concentrations of cisplatin (Figure 10G). Accordingly, following RRM2 knockdown, the IC50 values of cisplatin significantly decreased in T24 and UMUC3 cells (Figure 10H), implying that RRM2 may cause cisplatin resistance in BLCA. Colony assays exhibited that inhibiting RRM2 would reduce the cloning numbers of T24 and UMUC3 cells (Figure 10I)**.** These findings indicate that RRM2 plays an oncogenic role in BLCA. 

## 4. Discussion

Over the past decades, we have witnessed a rapidly advanced and evolved understanding of human cancers. Recent advances in high-throughput sequencing for tumor genomes have revolutionized cancer research studies and provided a convenient approach to precision medicine [25,26]. Pan-cancer analysis is applied to research TME signatures during vital pathophysiology processes of tumors [27]. Therefore, although the initial understanding of the role of RRM2 involves DNA synthesis, the potential carcinogenic effect of RRM2 is worth more research and exploration. In this study, we elucidated the comprehensive landscape of RRM2 in human cancers. With different bioinformatics platforms, we investigated the expression pattern, prognostic significance, mutational landscapes, gene interaction network, signaling pathways, immune infiltration, and predicted role in immunotherapy of RRM2 in tumors. We hope to provide a new perspective for understanding RRM2 in cancers and new strategies for tumor immunotherapy.

Previous studies concurred that increased RRM2 activity was tightly associated with tumor progression and malignancy [18,28]. For example, RRM2 elevated expression in mice markedly cooperated with activated oncogenes and drove malignancy [29]. Zhang et al. reported that RRM2 could promote CRC metastasis and invasion [14]. In accordance with these studies, we found that RRM2 expression in tumor tissues was increased compared to that in normal tissues across pan-cancer (Figure 1). High expression of RRM2 may lead to advanced pathologic stage across TCGA cancers (Figure 2B). Besides, RRM2 had a good prediction accuracy of diagnosis (AUC > 0.90) in pan-cancer (Appendix A). Furthermore, Cox regression analyses and KM survival curves demonstrated that RRM2 predicted poor OS and DSS in most cancers with TCGA and GEO (Figure 3 and Figure 4). Likewise, PFS and DFS analysis also showed that RRM2 was an unfavorable factor for tumor patients (Appendix A). New insights into RRM2 might enlighten and expand the thinking of clinical diagnosis and treatment with RRM2 inhibitors. Additionally, RRM2 inhibitors have been applied to treat solid tumors and blood malignant tumors as a single agent or in combination with other therapies [30,31]. In future updates, we hope that the RRM2 inhibitor could serve as a new combination agent of therapy target in cancers.

PPI analysis indicated that RRM2 and its cooperated genes were primarily related to the cell cycle, p53 signaling pathway, GSH metabolism, and drug metabolism processes (Figure 5F), implying RRM2 may be involved in ferroptosis and drug resistance across cancers. Consistent with this, Zhou et al. found a small-molecule blocking RRM2 could inhibit epidermal cancer cell growth and overcome hydroxyurea and gemcitabine resistance [32]. Markowitsch and others suggested that induction of ferroptosis might prevent KIRC development and improve the sensitivity of Sunitinib [33]. GSEA analysis suggested that RRM2-related genes were abundant in immune-related signals (B cell and BCR and complement system) and hallmarks of cancer processes (MAPK and NF-KB activation) in BRCA, LAML, LIHC, KIRC, KIRP and PRAD (Figure 6). These results suggest that RRM2 is strongly associated with TIME and malignant tumor hallmarks. 

Furthermore, we estimated the immune compartments of cancers, and discovered that RRM2 was correlated with the infiltrating immune cells. A significant association between RRM2 and most immune cells was observed across TCGA cancers (Figure 7A). Stromal cells, tumor cells and infiltrating immune cells constitute the bulk of the TIME, a critical factor of tumor biology [34,35]. As the major components in TIME, immune infiltrating cells master tumor suppression and immune escape [36]. Oncogenes can rebuild TIME and inhibit antitumor immunity via interacting with immune cells or stromal cells [37,38]. In line with our findings, it was reported that RRM2 facilitates tumor immune infiltration by inhibiting ferroptosis in LUAD [39]. In KIRC, the strong overexpression of RRM2 was correlated with T cell infiltration [40]. Continually, our preliminary study described that RRM2 expression was related to the decreased infiltration of NKT cells, implicating the latent impact of RRM2 on tumorigenesis (Figure 7C). Ample evidence has supported that NKT cells constitute a unique subset of T cells. NKT cells release inflammatory cytokines and modulate the function of effectors and regulatory immune cells, thus impacting antitumor immunity [41]. Past literature pointed out that NKT cells were considered as an interesting target for immunotherapy in CRC [42]. Next, we disclosed a close relationship between RRM2 and immune-regulated genes (Figure 9). It is prudent to explore the correlation between RRM2 and immune with a complete understanding of the mechanism of RRM2. 

Another important finding in this study is that that RRM2 high levels might predict a good response to immunotherapy (Figure 8A). We presented a strong link between RRM2 and ICI molecules in BLCA, BRCA, HNSC, KIRC, LGG, LIHC, LUAD, OV, THCA and UVM. ICI therapy is an encouraging treatment for human tumors [43]. Unfortunately, only a fraction of patients respond, presumably due to inadequate immune activation [44]. Hence, it becomes quite necessary to explore additional promising treatments which can collaborate with ICIs. In agreement with our results, Xiong et al. have documented that blocking RRM2 could enhance the antitumor efficiency of PD-1 blockade in renal cancer [45]. Subsequently, we disclosed that RRM2 expression was also correlated with TMB, MSI, NEO and CTL in most cancers, predicting the benefits of immunotherapy (Figure 8). TMB, MSI, NEO, and CTL emerge as effective immune-related therapeutic targets across cancers. High TMB means that more tumor neoantigens are exposed, so high TMB consistently selects for the benefit of ICIs blockade therapy [46]. MSI manifested DNA-mismatch repair-deficiency and it is a marker for a good response to immunotherapy [47]. NEO is mainly a tumor-specific antigen generated by mutations and is only expressed in tumor cells, which is considered a breakthrough in immunotherapy [48]. CTL expresses the CD8 coreceptor and is the preferred immune cell for killing cancer cells, so the density of CTL infiltrated is also a predictor for evaluating immunotherapy outcomes [49]. Briefly, our current research delineates the relationships between RRM2 and immunotherapy strategies, which could be instructive for clinicians. The role of RRM2 in immunology is complex and even contradictory across different cancers, and further studies are awaited.

Lastly, we validated RRM2 expression and function in BLCA. As the ninth most frequently diagnosed cancer worldwide, BLCA poses huge health hazards to society [1]. As urologists, we focus on urologic neoplasms. Regarding PRAD, it did not achieve statistical significance in bioinformatics analysis. Meanwhile, the function of RRM2 in renal cancer has been validated, so we did not select renal cancer to conduct further validation. As regards BLCA, we found that RRM2 showed a significantly differential expression and predicted poor survival of BLCA in bioinformatics analysis. RRM2 was related to immune infiltration and immunotherapy in the bioinformatics analysis of BLCA, so we assumed that RRM2 might function in BLCA. Besides, immunotherapy has been the first-line treatment for BLCA patients who are unable to tolerate cisplatin chemotherapy. However, the biomarker for BLCA immunotherapy is still lacking. Thus, BLCA is selected for experimental analysis. Literature reported that RRM2 was a novel diagnostic marker and a potential therapeutic target in BLCA, but they just justified this result in a tissue microarray with IHC and conducted a single proliferation experiment in one BLCA cell line UMUC3 [50]. Meanwhile, another two studies suggested that blocking RRM2 could enhance BLCA cells’ sensitivity to gemcitabine [51,52]. Conversely, we determined that RRM2 also could enhance BLCA cells’ sensitivity to cisplatin. We provided promising applications of RRM2 in emerging immunotherapy. In line with their results, we uncovered that RRM2 was indeed overexpressed in BLCA cell lines and tumor tissues with qRT-PCR, indicating that RRM2 may exert an oncogenic function in BLCA (Figure 10A,B). More importantly, we justified that blocking RRM2 with siRNA could significantly suppress BLCA cancer cell growth with CCK-8 and colony assays (Figure 10F–I). This provided further evidence that RRM2 might promote the progress and development of BLCA. Finally, we substantiated that RRM2 could cause cisplatin resistance in BLCA, suggesting that targeting RRM2 may enhance chemotherapy sensitivity in BLCA patients (Figure 10G–H). These results validate our bioinformatics analysis results, unearth the cancer-promoting role of RRM2, and prompt the therapeutic values of RRM2 in BLCA. 

This article suggested that RRM2 may influence TME to regulate tumor progression. We systematically analyzed RRM2 in human cancers. The innovations of our article lay in the following aspects. At first, we centered on the hotspots of ferroptosis-related gene RRM2 and immunotherapy. Next, we explored the correlation between RRM2 and TME. In particular, we discovered that RRM2 was correlated with predictors of immunotherapy, which might provide a reference for guiding immunotherapy in cancers. Lastly, laboratory verification was conducted in BLCA with our clinical samples and cancer cells. 

However, there are limitations in this study. Firstly, our study is just a fishing expedition that derives mainly from the computational analysis of genomic data. The exact mechanism of RRM2 in cancers still needs to be tested in vivo and in vitro. Secondly, our study only conducted experiments in BLCA, and the specific role of RRM2 in diverse cancers remains to be elucidated. Further cellular mechanism experiments and animal experiments are wanted. Thirdly, our study has the limitation of the biological functions of RRM2 in connection with immunotherapy. For further research, it is worth co-culturing tumor cells with either autologous or non-autologous immune cells, with LDH release, flow cytometry, real-time imaging or cytokine release assays using the interference of RRM2. The role of RRM2 in response to immunotherapy should be validated in clinical cancer patients. 

## 5. Conclusions

In conclusion, we preliminarily estimated the expression and the prognostic value of RRM2 in pan-cancer with bioinformatics prediction and experimental validation. Further, we found that RRM2 might be a future biomarker and a reference to predict immune response. These findings may aid in understanding the role of RRM2 and its clinical application in cancers. 

## Figures and Tables

**Figure 1 cancers-14-02938-f001:**
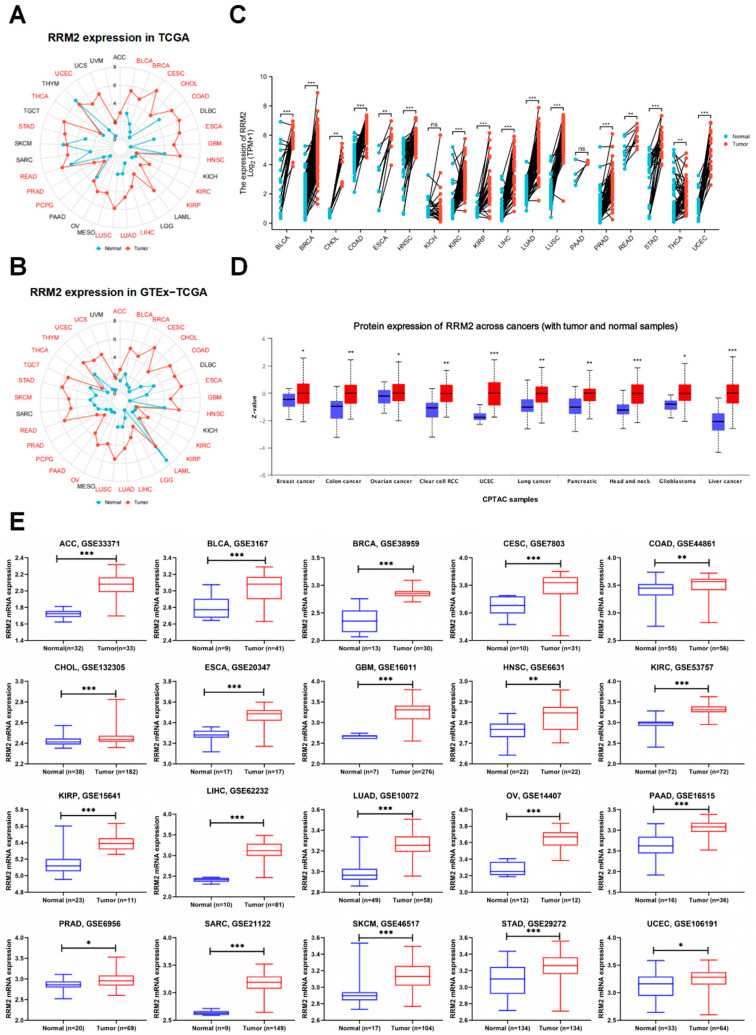
Overexpression of RRM2 in BLCA. (**A**,**B**) RRM2 mRNA expression in pan-cancer based on TCGA (**A**) and GTEx−TCGA database (**B**) (Red color indicated statistical significance). (**C**) RRM2 mRNA expression in paired tumor and adjacent normal tissues in TCGA database. (**D**) RRM2 protein expression in pan-cancer obtained from UALCAN database (http://ualcan.path.uab.edu/, accessed on 25 February 2022). (**E**) RRM2 mRNA expression in pan-cancer based on GEO database. (* *p* < 0.05, ** *p* < 0.01, *** *p* < 0.001).

**Figure 2 cancers-14-02938-f002:**
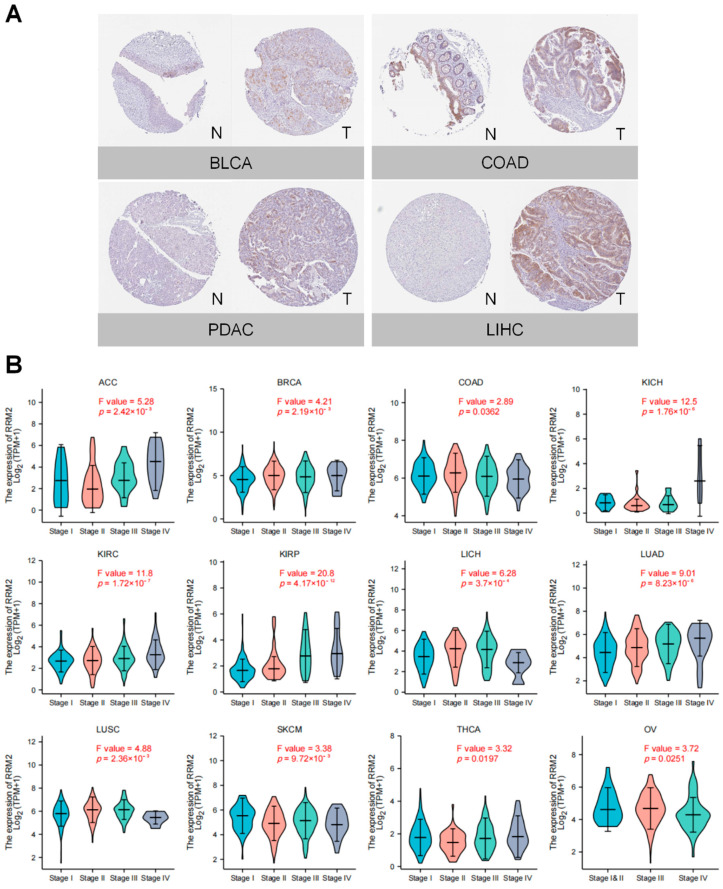
RRM2 Expression and clinical Features. (**A**) IHC images of RRM2 in pan-cancer. The figures were obtained from HPA database (https://www.proteinatlas.org/, accessed on 25 February 2022). (**B**) Correlation between RRM2 expression and tumor stages in pan-cancer analyzed in TCGA database.

**Figure 3 cancers-14-02938-f003:**
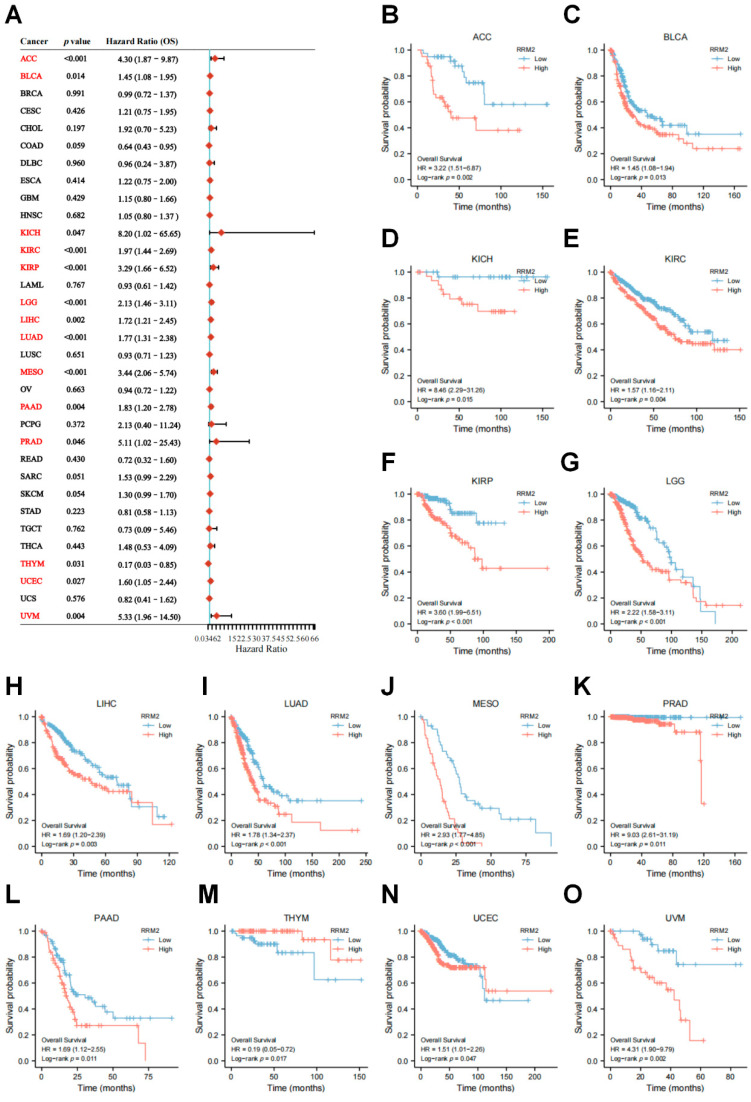
Relationship of RRM2 expression with Overall Survival (OS). (**A**) Forest map shows the univariate Cox regression analysis results for RRM2 in TCGA pan-cancer samples (Red color indicated statistical significance). (**B**–**O**) Kaplan–Meier OS curves of RRM2 expression in the significantly associated tumors.

**Figure 4 cancers-14-02938-f004:**
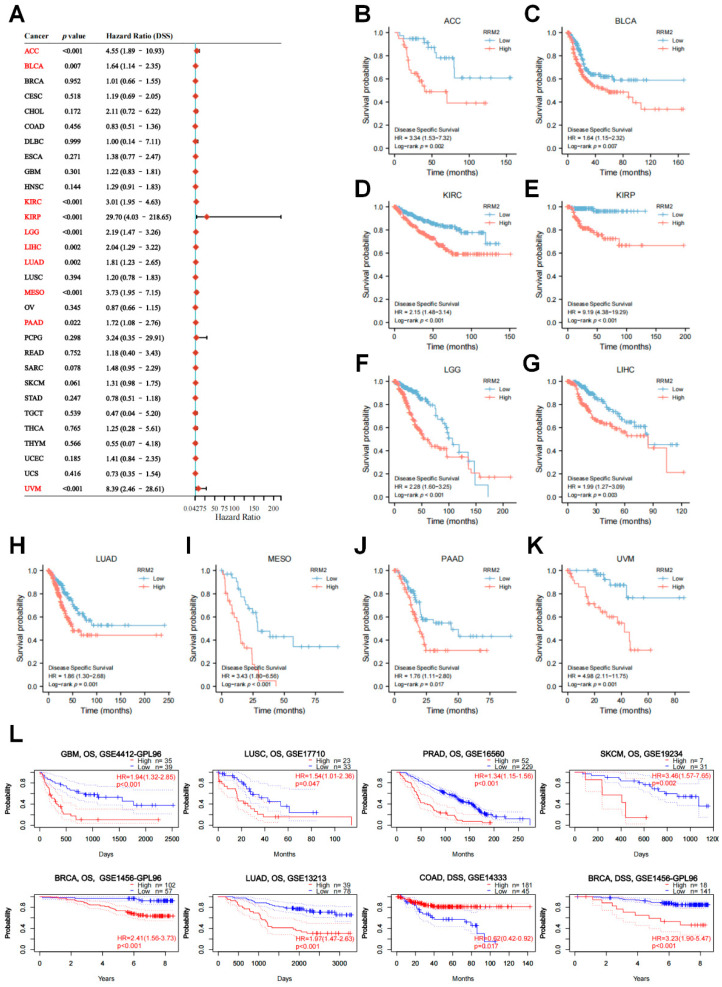
Relationship of RRM2 expression with Disease-Specific Survival (DSS). (**A**) Forest map shows the univariate Cox regression analysis results for RRM2 in TCGA pan-cancer samples (Red color indicated statistical significance). (**B**–**K**) Kaplan–Meier DSS curves of RRM2 expression in the significantly associated tumors. (**L**) The prognostic value of RRM2 in GEO with PrognoScan database.

**Figure 5 cancers-14-02938-f005:**
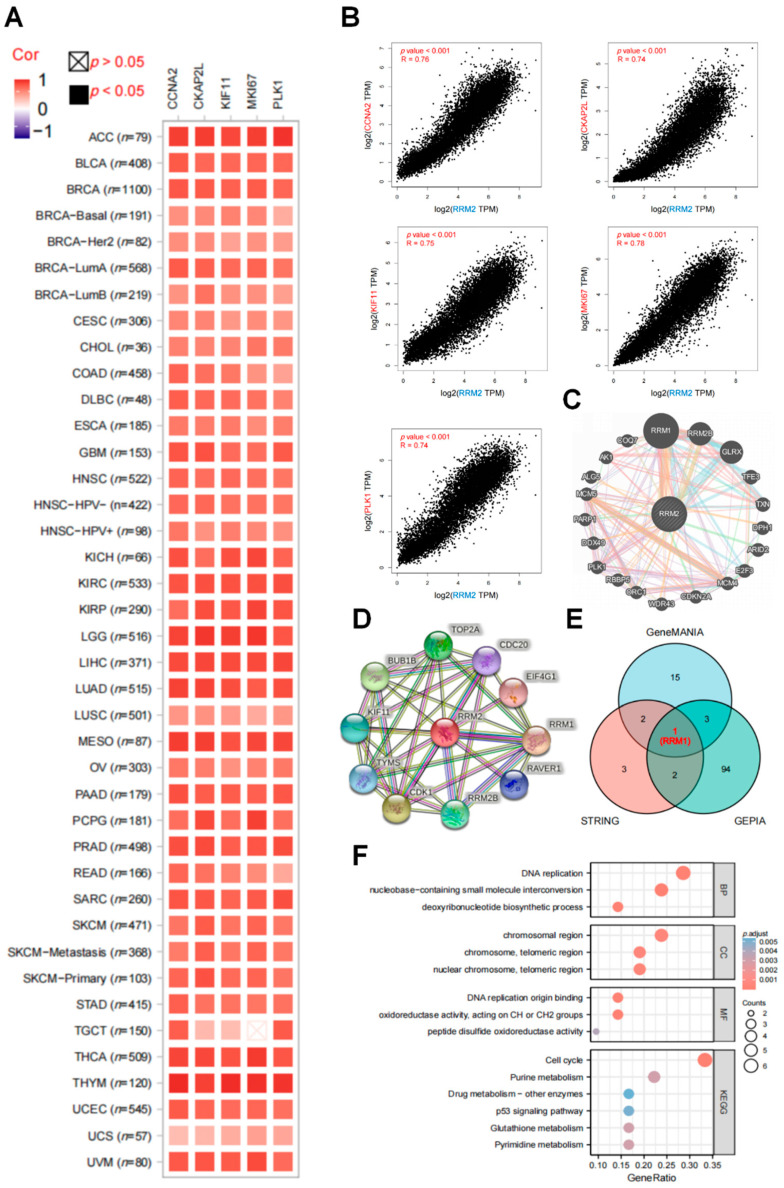
Enrichment analysis of RRM2-related partners. (**A**) The heatmap showed that RRM2 was positively related to the 5 selected genes in TCGA cancers via TIMER2.0. (**B**) The expression correlation between RRM2 and 5 selected genes (CCNA2, CKAP2L, KIF11, MKI67 and PLK1) (**C**,**D**) PPI network for RRM2 in GeneMANIA (http://www.genemania.org/, accessed on 26 February 2022) (**C**) and STRING (https://string-db.org/, accessed on 26 February 2022) (**D**). (**E**) Venn diagram of RRM2-interacted and correlated genes (RRM1). (**F**) KEGG and GO analyses of RRM2 to determine the functions and pathways of RRM2.

**Figure 6 cancers-14-02938-f006:**
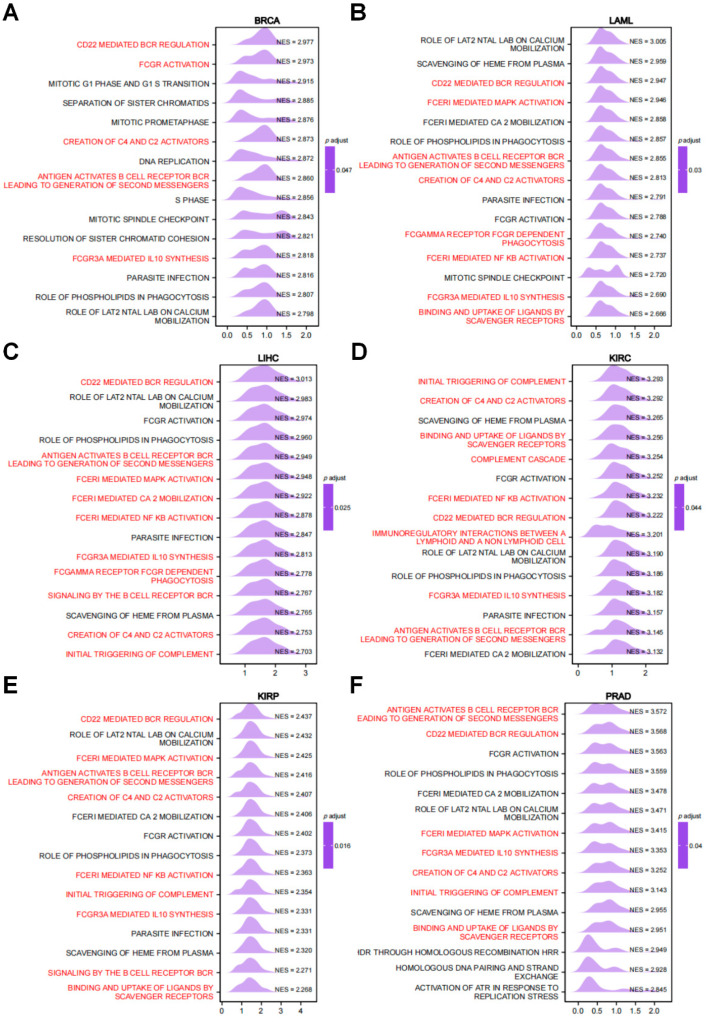
GSEA of RRM2 in pan-cancer. (**A**–**F**) The top 15 significant pathways of RRM2 GSEA analysis results across the indicated tumor types (Red color represents immune-related pathways).

**Figure 7 cancers-14-02938-f007:**
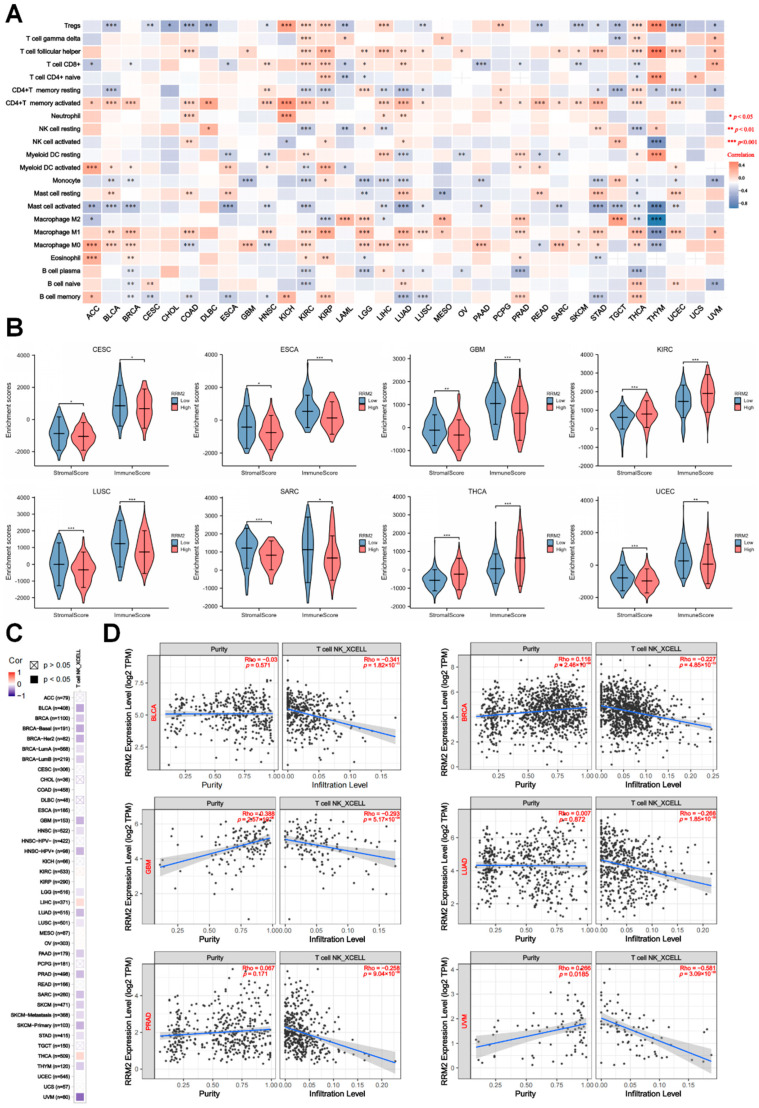
Correlation between RRM2 and immune infiltration in pan-cancer. (**A**) Correlations between RRM2 expression level and immune cells in pan-cancer. (**B**) Correlation between RRM2 expression and immune and stromal scores in multiple cancers. (**C**,**D**) Correlations between RRM2 expression level and the infiltration level of NK T cells across TCGA cancers. Heatmap (**C**) and scatter plots (**D**) of NK T cells infiltration level related to RRM2 expression were presented, utilizing the TIMER2.0 database. (* *p* < 0.05, ** *p* < 0.01, *** *p* < 0.001).

**Figure 8 cancers-14-02938-f008:**
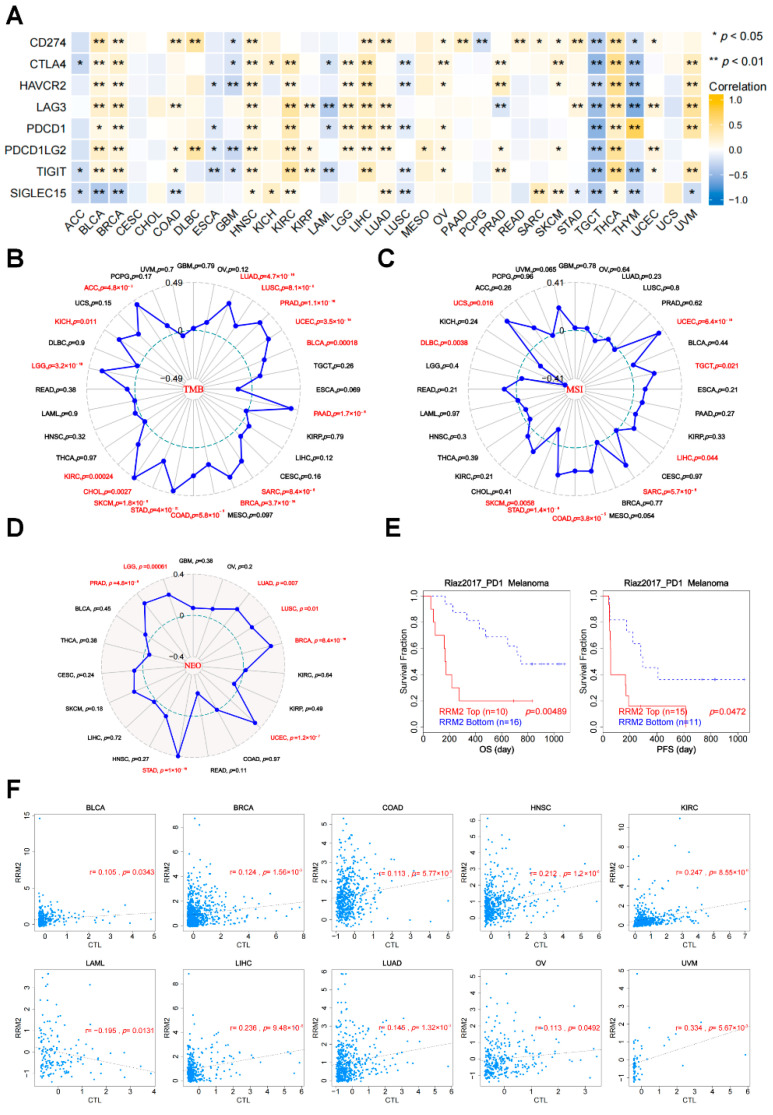
Correlation between RRM2 and immunotherapy. (**A**) Correlation of RRM2 expression with immune checkpoints. (**B**−**D**) Correlation of RRM2 expression with TMB (**B**), MSI (**C**) and NEO (**D**) (Red color indicated statistical significance). (**E**) Kaplan-Meier curves of survival ratios as a measure of the PD1 immunotherapeutic response between high and low expression of RRM2 in in Riza2017_PD1 clinical study of melanoma. (**F**) The correlation between the RRM2 expression and CTL in TCGA cohorts. (* *p* < 0.05, ** *p* < 0.01, red color indicated statistical significance).

**Figure 9 cancers-14-02938-f009:**
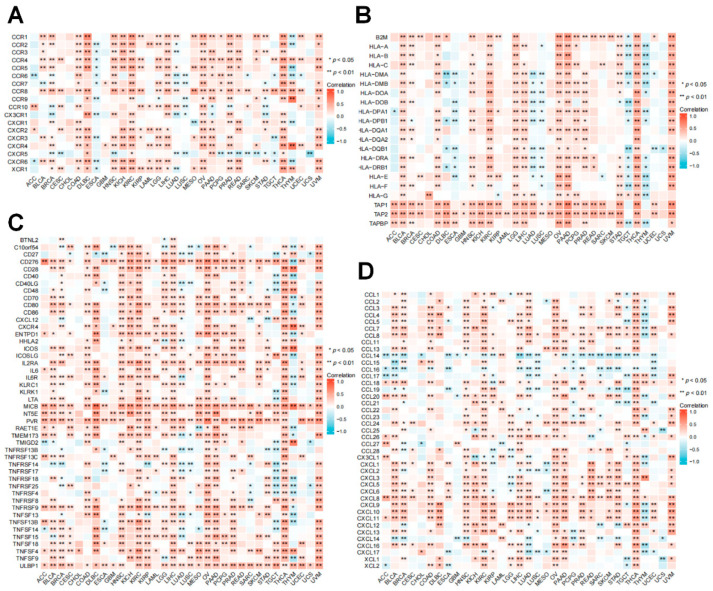
Correlation between RRM2 and immune-related genes. (**A**–**D**) Correlation between RRM2 and chemokine receptors (**A**), MHC genes (**B**), immune activated genes (**C**) and chemokines (**D**). (* *p* < 0.05, ** *p* < 0.01).

**Figure 10 cancers-14-02938-f010:**
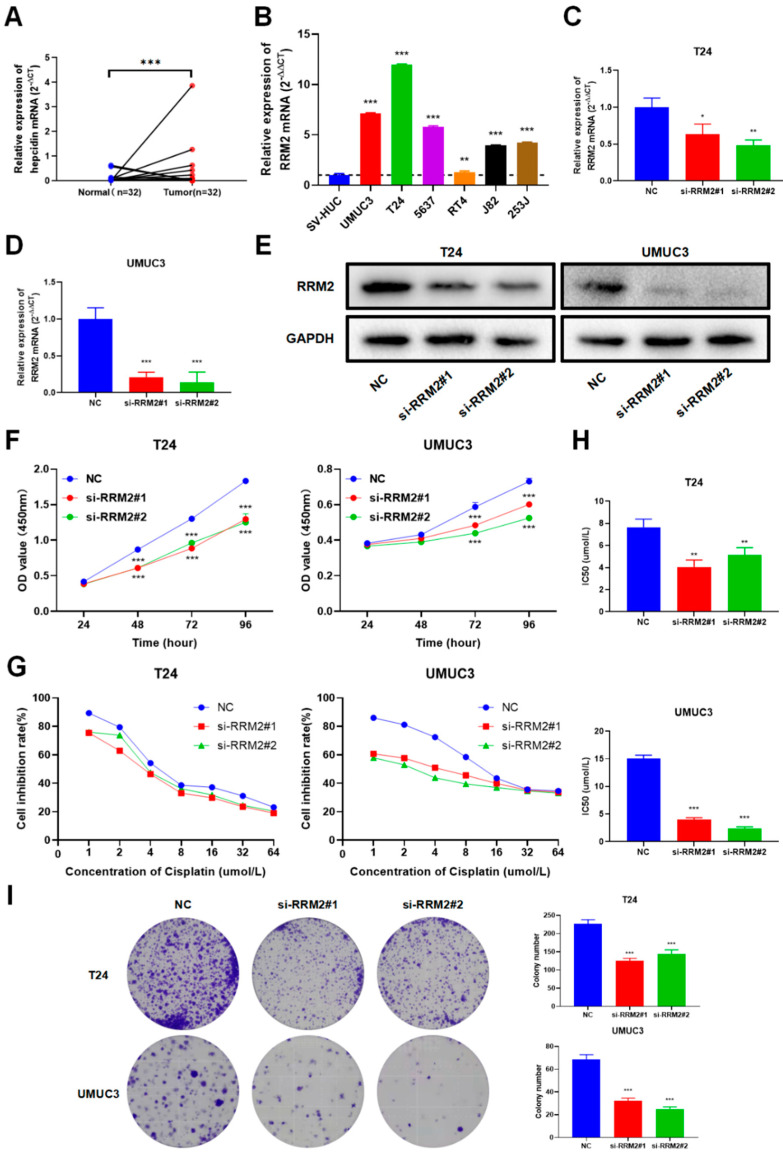
Expression and function of RRM2 in BLCA. (**A**,**B**) RRM2 mRNA expression in BLCA paired tissues (**A**) and cell lines (**B**). (**C**–**E**) RRM2 siRNA transfection efficiency was detected by qRT-PCR (**C**,**D**) and western blot (**E**). (**F**) T24 and UMUC3 cells’ growth and proliferation was explored by CCK-8 assays. (**G**,**H**) Down-expression of RRM2 could enhance the sensitivity of T24 and UMUC3 cells to cisplatin (**G**) and the bar graphs of IC50 were presented (**H**). (**I**) Blocking RRM2 expression could decrease the colony number of T24 and UMUC3 cells. (* *p* < 0.05, ** *p* < 0.01, *** *p* < 0.001).

**Table 1 cancers-14-02938-t001:** Correlation between RRM2 expression and clinicopathological features in TCGA-BLCA patients (*n* = 539).

Characteristics	Odds Ratio (OR)	*p* Value
Age (>70 vs. ≤70)	0.854 (0.579–1.260)	0.428
Gender (Male vs. Female)	0.928 (0.598–1.438)	0.738
T stage (T3&T4 vs. T1&T2)	1.211 (0.789–1.864)	0.382
N stage (N1&N2&N3 vs. N0)	1.088 (0.711–1.669)	0.697
M stage (M1 vs. M0)	1.493 (0.436–5.332)	0.519
Pathologic stage (Stage III&Stage IV vs. Stage I&Stage II)	1.305 (0.864–1.976)	0.207
Histologic grade (High Grade vs. Low Grade)	6.447 (2.140–27.837)	**0.003 ****

** *p* < 0.01.

**Table 2 cancers-14-02938-t002:** Multivariable Cox regression of RRM2 and clinical features of BLCA in TCGA.

Characteristics	OS		DSS	
	HR (95% CI)	*p* Value	HR (95% CI)	*p* Value
Age (>70 vs. ≤70)	1.250 (0.780–2.002)	0.354	--	
Gender (Male/Female)	--		--	
T stage (T3&T4/T1&T2)	3.066 (1.045–8.995)	**0.041 ***	3.018 (0.872–10.445)	0.081
N stage (N1&N2&N3/N0)	2.139 (1.255–3.646)	**0.005 ****	2.777 (1.417–5.444)	**0.003 ****
M stage (M1/M0)	1.317 (0.504–3.442)	0.574	1.453 (0.489–4.316)	0.501
Pathologic stage (Stage III&IV/Stage I&II)	0.511 (0.157–1.663)	0.265	0.520 (0.125–2.155)	0.367
Histologic grade (High Grade/Low Grade)	--		--	
RRM2 (High Expression/Low Expression)	1.693 (1.052–2.723)	**0.030 ***	2.482 (1.331–4.628)	**0.004 ****

* *p* < 0.05, ** *p* < 0.01.

## Data Availability

Data presented in the paper is available upon request from corresponding author.

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
