# Peer review of "Comprehensive Landscape of RRM2 with Immune Infiltration in Pan-Cancer"

_cancers, 2022, doi:10.3390/cancers14122938_

Round 1

Reviewer 1 Report

Zhou et al. focused the gene RRM2 and analysed the correlation with the progression of pan-cancer by using bioinformatics analysis, they predicted the network by in silio analusis and demonstrated the consistence by experiments.

Although the reviewer acknowledges the scientific significance of this research object, the manuscript and the figures are a mixture of the citation of the database and their original experimental data, and it is difficult to evaluate what efforts were made by the authors. For instance, Figure 1, what dataset (database?) did the authors use? The methods of the data process have not been described in the text and in the figure legends. If they described the methods in the methods section, the correspondence of the methods and the data must be clearly written. Figure 2A is the image capture of the public database (I believe). Although this is lightly mentioned at the figure legends, the authors have to emphasise the source of the data by citing the link of the website etc. and declare that the data have been taken somewhere else. Moreover, as I believe, the image captures cannot be used in the original research article. Figure 5C,D have the same problem as in Figure 2A.

They mentioned natural killer and immune infiltration, but they only use the bladder cancer cells in the expriments. The rationale must be provided.

Text;

Line 15; In this article, we provided a comprehensive analysis of RRM2 and immune infiltration in pan-cancer. Please check the sentence. Do they mean “.. and demonstrate the correlation with the immune infiltration in pan-cancer”

Line 93: All patients: how many patients?

Line 97; T24, J82, … which are SVHUC and which are BLCA cell lines?

Line 175: Up-expression; I don’t think this is a usual English expression.

Line 204: RRM2 upregulation leads to advanced tumor stages; however, in Fig 2B, the tendency is not evident.

Line 258: subheadings (RRM2 alteration analysis) must reflect the main claim of the part. What is the main claim of this part?

Line 266: the same above; what have been found from the network analysis?

Line 292, 303, 330; the same as above.

The discussion must be written. They describe the experimental results in the discussion section without any interpretation. This statement must be made in the results part (e.g. line 497-501).

Author Response

Dear Editor/Reviewers:

Thank you very much for your prompt attention to our manuscript and helpful suggestions. We have resubmitted our revised manuscript entitled “Comprehensive Landscape of RRM2 with Immune Infiltration in Pan-cancer” to the “Cancers” for your reconsideration of its suitability for publication (Manuscript ID: cancers-1745370). All authors have read and approved the comments. We have carefully taken your comments into account and responded to each of the points raised by the reviewers. All the altered passages have been highlighted in red. The corresponding explanations of each point which is raised in your comments are as follows:

Point 1: Although the reviewer acknowledges the scientific significance of this research object, the manuscript and the figures are a mixture of the citation of the database and their original experimental data, and it is difficult to evaluate what efforts were made by the authors. For instance, Figure 1, what dataset (database?) did the authors use? The methods of the data process have not been described in the text and in the figure legends. If they described the methods in the methods section, the correspondence of the methods and the data must be clearly written.

Response: Thanks for your careful review and insightful comments. Generally, all the experimental data were presented in Figure 10 while the results of bioinformatics analysis were shown in Figures 1-9. For instance, Figure 1, we use the data from TCGA and GTEx database to conduct differential RRM2 mRNA expression analysis of normal and tumor samples, using the “beeswarm” and “ggpubr” R packages. Expression data were Log2 transformed and two sets of t-tests were conducted on these tumor types; P < 0.05 indicated differential expression between tumor and normal tissues. We have added the special and clear methods of each figure in “Materials and Methods” section and in figure legends (The red in line 114-171 and in figure legends).

Figure 2A is the image capture of the public database (I believe). Although this is lightly mentioned at the figure legends, the authors have to emphasise the source of the data by citing the link of the website etc. and declare that the data have been taken somewhere else. Moreover, as I believe, the image captures cannot be used in the original research article. Figure 5C,D have the same problem as in Figure 2A.

Response: Figure 2A and Figure 5C, D are indeed the image capture of the public database. We have added the source of the data by citing the link of the website and added the declaration in “Materials and Methods” and in figure legends (Line 233-236, 314-317 ). The image captures used in the original research seem acceptable in the bioinformatics analyses articles that have been published in Cancers [1-4]. However, we agree with your remarks and we are willing to put these images into the supplementary materials.

  1. Izzi, V.; Davis, M.N.; Naba, A. Pan-Cancer Analysis of the Genomic Alterations and Mutations of the Matrisome. Cancers 2020, 12, doi:10.3390/cancers12082046.
  2. Vetrivel, P.; Murugesan, R.; Bhosale, P.B.; Ha, S.E.; Kim, H.H.; Heo, J.D.; Kim, G.S. A Network Pharmacological Approach to Reveal the Pharmacological Targets and Its Associated Biological Mechanisms of Prunetin-5-O-Glucoside against Gastric Cancer. Cancers 2021, 13, doi:10.3390/cancers13081918.
  3. van Wilpe, S.; Wosika, V.; Ciarloni, L.; Hosseinian Ehrensberger, S.; Jeitziner, R.; Angelino, P.; Duiveman-de Boer, T.; Koornstra, R.H.T.; de Vries, I.J.M.; Gerritsen, W.R.; et al. Whole Blood Transcriptome Profiling Identifies DNA Replication and Cell Cycle Regulation as Early Marker of Response to Anti-PD-1 in Patients with Urothelial Cancer. Cancers 2021, 13, doi:10.3390/cancers13184660.
  4. Yang, J.L.; Wang, C.C.N.; Cai, J.H.; Chou, C.Y.; Lin, Y.C.; Hung, C.C. Identification of GSN and LAMC2 as Key Prognostic Genes of Bladder Cancer by Integrated Bioinformatics Analysis. Cancers 2020, 12, doi:10.3390/cancers12071809.

They mentioned natural killer and immune infiltration, but they only use the bladder cancer cells in the experiments. The rationale must be provided.

Response: Thank you for your insightful question. Our study does exist the limitation that we only used the bladder cancer cells in the experiments. We are sorry that we may not validate RRM2 function in multiple types of cancers due to the limitation of experimental conditions. As urologists, we focus on urologic neoplasms, including bladder cancer (BLCA), prostate adenocarcinoma (PRAD), and renal cancer (including KIRC, KIRP, and KICH). Regarding PRAD, it did not achieve statistical significance in bioinformatics analysis. Meanwhile, the function of RRM2 in renal cancer has been validated [1-3], so we did not select renal cancer to conduct further validation. As regards BLCA, we found that RRM2 showed a significantly differential expression (Figure 1-2) and predicted poor survival (Figure 3-4) of BLCA in bioinformatics analysis. Also, RRM2 was related to immune infiltration (Figure 7) and immunotherapy (Figure 8) in the bioinformatics analysis of BLCA, so we assumed that RRM2 might function in BLCA. Although the concept and technology of diagnosis and treatment of BLCA have developed rapidly, the biomarker of BLCA is still lacking. Thus, BLCA is selected for experimental analysis. We have added this motivation for why BLCA is selected in “Discussion” (Line 520-531).

Also, our study does have a limitation on the validation of the biological functions of RRM2 in connection with the natural killer and immune infiltration. Our exploratory study was just a hypothesis expedition. However, we are collecting clinical data of bladder cancer patients who response or did not response to immunotherapy for further analysis. In the future, we will try to co-culture tumor cells with either autologous or non-autologous immune cells for monitoring immune-mediated tumor killing, with LDH release, flow cytometry, real-time imaging, or cytokine release assays using the interference for RRM2. Thank you again for your advice that indicates the direction for our future research. We have added this description of this limitation in “Discussion”, line 556-562.

  1. Szarkowska, J.; Cwiek, P.; Szymanski, M.; Rusetska, N.; Jancewicz, I.; Stachowiak, M.; Swiatek, M.; Luba, M.; Konopinski, R.; Kubala, S.; et al. gene expression depends on BAF180 subunit of SWISNF chromatin remodeling complex and correlates with abundance of tumor infiltrating lymphocytes in ccRCC. Am J Cancer Res 2021, 11, 5965-5978.
  2. Xiong, W.; Zhang, B.; Yu, H.; Zhu, L.; Yi, L.; Jin, X. RRM2 Regulates Sensitivity to Sunitinib and PD-1 Blockade in Renal Cancer by Stabilizing ANXA1 and Activating the AKT Pathway. Advanced science (Weinheim, Baden-Wurttemberg, Germany) 2021, 8, e2100881, doi:10.1002/advs.202100881.
  3. Zou, Y.; Zhou, J.; Xu, B.; Li, W.; Wang, Z. Ribonucleotide reductase subunit M2 as a novel target for clear-cell renal cell carcinoma. OncoTargets and therapy 2019, 12, 3267-3275, doi:10.2147/ott.S196347.

Point 2: Text;

Line 15; In this article, we provided a comprehensive analysis of RRM2 and immune infiltration in pan-cancer. Please check the sentence. Do they mean “.. and demonstrate the correlation with the immune infiltration in pan-cancer”

Response: We are really sorry that we did not state it clearly. It means “we provided a comprehensive analysis of RRM2 with immune infiltration in pan-cancer.” We have modified the sentence in our paper (Line 14).

Line 93: All patients: how many patients?

Response: A total of 32 bladder cancer patients were included in our research. A total of 32 matched bladder tumor and adjacent normal specimens were obtained by radical nephrectomy in our center. We have added the explanation (Line 88-92).

Line 97; T24, J82, … which are SVHUC and which are BLCA cell lines?

Response: T24, J82, UMUC3, RT4, 5637, and 253J are all BLCA cell lines. Simian Virus 40-immortalized human uroepithelial cell (SVHUC) is the human normal urinary epithelial cell line, which is often used as a normal control [1]. We have added the explanation (Line 94-97).

  1. Christian, B.J.; Kao, C.H.; Wu, S.Q.; Meisner, L.F.; Reznikoff, C.A. EJ/ras neoplastic transformation of simian virus 40-immortalized human uroepithelial cells: a rare event. Cancer research 1990, 50, 4779-4786.

Line 175: Up-expression; I don’t think this is a usual English expression.

Response: Thank you for your careful review. We have corrected it to “Overexpression” (Line 196).

Line 204: RRM2 upregulation leads to advanced tumor stages; however, in Fig 2B, the tendency is not evident.

Response: Thank you for your correction. More precisely, RRM2 expression was significantly related to tumor stages. Moreover, RRM2 upregulation leads to advanced tumor stages in several cancers, including KIRC, KIRP and LUAD. We have made corrections in our paper (Line 226-231).

Line 258: subheadings (RRM2 alteration analysis) must reflect the main claim of the part. What is the main claim of this part?

Response: We highly approve of your comments. The accumulation of genetic alterations drives the progression of normal cells through hyperplastic and dysplastic stages to invasive cancer. Gene alteration analyses should therefore provide further insights into cancer progression. Therefore, we assessed them for RRM2. This section (RRM2 alteration analysis) aimed to present the genetic alteration and mutation landscape of RRM2 in cancers and we have modified the subheadings (Line 281-283, 291-293).

Line 266: the same above; what have been found from the network analysis?

Response: The network analysis aimed to find the proteins that interacted with RRM2. Moreover, it aimed to find further the potential pathways that RRM2 was implicated in by GO, KEGG and GSEA analysis. This section found that RRM2 was involved in tumor progression and immune-related signaling pathways in human cancers. (Line 294, 306-310, 319-324)

Line 292, 303, 330; the same as above.

Response: We have modified the subheadings you mentioned to reflect the main claim of the corresponding part and summarized the main claim of each part at the end of each section (Line 328, 346-348, 356, 375-376).

The discussion must be written. They describe the experimental results in the discussion section without any interpretation. This statement must be made in the results part (e.g. line 497-501).

Response: We very much agree. We have modified the discussion to interpret our experimental results (Line 537-546). Also, we have improved our discussion (Line 445-450, 479-488, 498-505). Besides, the writing of our article has been modified and improved by a native English speaker. Thank you again for your support.

Lastly, we appreciate your work earnestly and hope that the correction will meet with approval. Please refer to the present manuscript text for details. We would be glad to further modify the text if you have other suggestions. Once again, special thanks to you for your suggestion.

Best regards

Zhong Wu

Department of Urology, Huashan Hospital, Fudan University

Shanghai, China

Reviewer 2 Report

Zhou et.al has performed pan-cancer analysis of RRM2 with immune infiltration. Authors approached it with bioinformatics analysis. It is an interesting study but I have following concerns

Major comments

1)   In section 3.1 authors checked expression of RRM2 variety of cancer datasets from TCGA, GTEX and paired non-cancer samples. How paired non-cancer samples were obtained.

2)   In 3.2 authors describe RRM2 correlation with tumor stages BLCA cancer dataset is missing. Authors need to explain about this dataset.

3)   They also performed prognostic analysis of RRM2 using cox regression analysis in TCGA. Validated using prognoscan and KM plotter tools why BLCA data analysis is missing in this section.

4)   In section 3.4 authors performed network analysis of RRM2 and acquired top 100 RRM2 associated genes were considered. How these top 100 were acquired and correlation analysis is performed using top 5 genes. It is not clear which cut-offs were used to select top 100 and 5 genes.

5)   In section 3.9 authors performed RRM2 expression is validated using series of experiments in BLCA. Motivation is not clear why BLCA is selected for the analysis as in bioinformatics analysis did not show BLCA in different analysis.

Reviewer 3 Report

The authors assessed the association of RRM2 and clinically relevant outcomes in several types of cancer with public databases. Also, the authors define the essential genes and signal pathways regulated by RRM2. The most relevant result was the relation of RRM2 with immune Cells.

This study has a limitation on the biological functions of RRM2 in connection with immunotherapy. A possible option is the Co-culture of tumor cells with either autologous or non-autologous immune cells for monitoring immune-mediated tumor killing, with standard readouts being HCI, LDH release, flow cytometry, real-time imaging, or cytokine release assays using the interference for RRM2

Author Response

Dear Editor/Reviewers:

Thank you very much for your prompt attention to our manuscript and helpful suggestions. We have resubmitted our revised manuscript entitled “Comprehensive Landscape of RRM2 with Immune Infiltration in Pan-cancer” to the Cancers for your reconsideration of its suitability for publication (Manuscript ID: cancers-1745370). All authors have read and approved the comments. We have carefully taken your comments into account and responded to each of the points raised by the reviewers. All the altered passages have been highlighted. The corresponding explanations of each point which is raised in your comments are as follows:

Point 1: The authors assessed the association of RRM2 and clinically relevant outcomes in several types of cancer with public databases. Also, the authors define the essential genes and signal pathways regulated by RRM2. The most relevant result was the relation of RRM2 with immune Cells.

This study has a limitation on the biological functions of RRM2 in connection with immunotherapy. A possible option is the Co-culture of tumor cells with either autologous or non-autologous immune cells for monitoring immune-mediated tumor killing, with standard readouts being HCI, LDH release, flow cytometry, real-time imaging, or cytokine release assays using the interference for RRM2.

Response 1: Thank you very much for the insightful suggestions. We fully agree with your remarks. Our study does have a limitation on the biological functions of RRM2 in connection with immunotherapy. Recently, we are trying to co-culture tumor cells with mast cells (HMC-1) or macrophages (THP-1) for further experiments. Besides, we are collecting clinical data of bladder cancer patients who receiving immunotherapy for further analysis. Also, the experiment methods you mentioned are quite instructive and interesting. In the future, we will try to conduct these assays as you advise. Thank you again for your advice that indicates the direction for our future research. We have added this description in Discussion, line 556-562.

Lastly, we appreciate your work earnestly and hope that the correction will meet with approval. Please refer to the present manuscript text for details. We would be glad to further modify the text if you have other suggestions. Once again, special thanks to you for your suggestion.

Best regards

Zhong Wu

Department of Urology, Huashan Hospital, Fudan University

Shanghai, China

Round 2

Reviewer 1 Report

The authors respond to all of the comments and I now support publication of this manuscript.

Reviewer 2 Report

Authors answered all my queries 

Reviewer 3 Report

The paper can be accepted without any further changes.